# Vascular Smooth Muscle Cell-Specific Progerin Expression Provokes Contractile Impairment in a Mouse Model of Hutchinson-Gilford Progeria Syndrome that Is Ameliorated by Nitrite Treatment

**DOI:** 10.3390/cells9030656

**Published:** 2020-03-08

**Authors:** Lara del Campo, Amanda Sánchez-López, Cristina González-Gómez, María Jesús Andrés-Manzano, Beatriz Dorado, Vicente Andrés

**Affiliations:** 1Laboratory of Molecular and Genetic Cardiovascular Pathophysiology, Vascular Pathophysiology Area, Centro Nacional de Investigaciones Cardiovasculares (CNIC), Melchor Fernández Almagro 3, 28029 Madrid, Spain; laradelcampo@hotmail.com (L.d.C.); amanda.sanchez.asl@gmail.com (A.S.-L.); cristina.gonzalez@externo.cnic.es (C.G.-G.); mjandres@cnic.es (M.J.A.-M.); beatrizjulia.dorado@cnic.es (B.D.); 2CIBER de Enfermedades Cardiovasculares (CIBERCV), Spain

**Keywords:** HGPS, progerin, VSMCs, ECs, vascular function, sodium nitrite

## Abstract

Cardiovascular disease (CVD) is the main cause of death worldwide, and aging is its leading risk factor. Aging is much accelerated in Hutchinson–Gilford progeria syndrome (HGPS), an ultra-rare genetic disorder provoked by the ubiquitous expression of a mutant protein called progerin. HGPS patients die in their teens, primarily due to cardiovascular complications. The primary causes of age-associated CVD are endothelial dysfunction and dysregulated vascular tone; however, their contribution to progerin-induced CVD remains poorly characterized. In the present study, we found that progeroid *Lmna^G609G^*^/*G609G*^ mice with ubiquitous progerin expression show both endothelial dysfunction and severe contractile impairment. To assess the relative contribution of specific vascular cell types to these anomalies, we examined *Lmna^LCS^*^/*LCS*^*Tie2Cre^tg^*^/+^ and *Lmna^LCS^*^/*LCS*^*Sm22αCre^tg^*^/+^ mice, which express progerin specifically in endothelial cells (ECs) and vascular smooth muscle cells (VSMCs), respectively. Whereas vessel contraction was impaired in mice with VSMC-specific progerin expression, we observed no endothelial dysfunction in mice with progerin expression restricted to VSMCs or ECs. Vascular tone regulation in progeroid mice was ameliorated by dietary sodium nitrite supplementation. Our results identify VSMCs as the main cell type causing contractile impairment in a mouse model of HGPS that is ameliorated by nitrite treatment.

## 1. Introduction

Cardiovascular disease (CVD) is the leading cause of death worldwide, in part due to progressive aging, the main CVD risk factor [1]. A number of additional factors have been identified that increase the risk of developing CVD, either acting alone or in combination (e.g., hypercholesterolemia, diabetes, sedentary lifestyle, and smoking) [2,3,4]. However, studies on the effects of age alone (the only factor we cannot modify or treat) remain scarce due to their high research costs associated with the necessary long-term resource investment and delays in collecting results. The study of premature aging syndromes characterized by accelerated CVD thus offers a unique opportunity to investigate age-dependent drivers of CVD in the absence of other confounding risk factors [5,6].

Hutchinson–Gilford progeria syndrome (HGPS, OMIM 176670) is a rare genetic disease characterized by accelerated aging and death in adolescence [5,6,7,8,9]. HGPS children have impaired postnatal growth, lipodystrophy, alopecia, pigmented and wrinkled skin, and skeletal dysplasia. They also develop generalized atherosclerosis, arterial stiffness and calcification, electrocardiographic abnormalities, and ventricular diastolic dysfunction and die prematurely at an average age of 14.6 years mainly due to myocardial infarction, stroke, or heart failure [7,10,11]. HGPS is caused by a heterozygous de novo point mutation in the *LMNA* gene, which encodes the nuclear proteins lamin A and C (A-type lamins) [12,13]. This synonymous mutation activates a cryptic splice donor site that removes 150 nucleotides from exon 11, causing the synthesis of progerin [12,13]. This truncated form of prelamin A is expressed ubiquitously and acts in a dominant-negative manner, causing alterations in many essential nuclear functions, including nuclear structure, gene transcription, signal transduction, DNA damage repair, chromatin organization, mechanosensing, and proliferation [14,15]. Aside from its role in HGPS, progerin is detectable at low levels in several tissues during normal aging, including atherosclerotic coronary arteries, suggesting a role in physiological aging [11,16,17,18,19].

Homozygous *Lmna*-null mice lacking A-type lamins develop to term without exhibiting overt anomalies, but they develop skeletal muscle dystrophy and dilated cardiomyopathy soon after birth and are all death by the eight week [20]. Interestingly, “*Lamin C-Stop*” mice (LCS) expressing Lamin C but lacking lamin A are apparently normal, but are slightly heavier and longer-lived than wild-type controls [21]. Over the last decade, several animal models of HGPS have been generated and characterized [6,22]. *Lmna^G609G^*^/*G609G*^ mice, which express ubiquously progerin and lamin C and lack lamin A, recapitulate the main clinical manifestations of human HGPS, such as difficulty to thrive, lipodystrophy, skeleton abnormalities, vascular calcification and stiffening, vascular smooth muscle cell (VSMC) loss, and premature death [23,24,25]. Moreover, ubiquitous or VSMC-specific progerin expression accelerates atherosclerosis in atherosclerosis-prone *Apolipoprotein e-*null mice (*Apoe*^−/−^), at least in part due to excessive endoplasmic reticulum stress and associated unfolded protein response [26,27].

Vasomotor function is modulated by a delicate balance between constriction and dilation, which are mainly controlled by luminal endothelial cells (ECs) and medial VSMCs [28,29,30]. The most common pathological signs of vascular aging are endothelial dysfunction and vessel stiffness, which can progress over the years to hypertension, cardiac and vessel overload, inflammation, fibrosis, atherosclerosis, and heart failure [29,30,31]. The stiffness of aged vessels correlates with changes in the mechanical properties of VSMCs, which lose their ability to contract when surrounded by an abnormally stiff extracellular matrix [32,33]. Vessel stiffness is also a pathological signature of HGPS [7,34], and progerin-expressing *Lmna^G609G^*^/*G609G*^ mice exhibit altered vascular structure characterized by stiffness and inward remodeling associated with VSMC degeneration and increased collagen deposition in the medial layer [25]. In the present study, we investigated whether endothelial dysfunction and defective regulation of vascular tone characterize progeroid *Lmna^G609G^*^/*G609G*^ mice. Moreover, to determine the relative contributions made by progerin-expressing ECs and VSMCs, we bred *Lmna^LCS^*^/*LCS*^
*Tie2Cre*^+/*tg*^ and *Lmna^LCS^*^/*LCS*^
*SM22αCre*^+/*tg*^ mice, which express progerin specifically in ECs and VSMCs respectively, but do not show any apparent signs of aging [25]. We also examined the potential benefit of treatment with nitrites, a physiologically important storage form of nitric oxide (NO) [35] that can partly revert vascular endothelial dysfunction and stiffness, oxidative stress, and inflammation in physiological aging [36] and ameliorates structural stiffening in progeroid mice [25].

## 2. Materials and Methods

### 2.1. Mice

All procedures with mice conformed to EU Directive 2010/63EU and Recommendation 2007/526/EC, enforced in Spanish law under Real Decreto 53/2013. Animal protocols were approved by the local ethics committees and the Animal Protection Area of the Comunidad Autónoma de Madrid (PROEX 135/14). Mice were housed at the CNIC pathogen free facility and sacrificed at 14–15 weeks of age.

Studies were carried out with males of the following genotypes (all on the C57BL/6J genetic background): *Lmna^G609G^*^/*G609G*^ mice with constitutive and ubiquitous progerin and lamin C expression and lack of lamin A, obtained by crossing heterozygous *Lmna^G609G^*^/+^ males and females [23]; *Lmna^LCS^*^/*LCS*^*SM22αCre^tg^*^/+^ mice with ubiquitous lamin C expression and VSMC-specific progerin expression, obtained by crossing *Lmna^LCS^*^/*LCS*^ mice [23] with *SM22αCre*^+/−^ mice (The Jackson Laboratory, Bar Harbor, ME, USA); *Lmna^LCS^*^/*LCS*^*Tie2Cre^tg^*^/+^ mice with ubiquituous expression of lamin C and EC-specific progerin expression, obtained by crossing *Lmna^LCS^*^/*LCS*^ mice with *Tie2-Cre*^+/−^ mice [37]. Controls for *Lmna^G609G^*^/*G609G*^ mice were littermate wild-type mice expressing normal lamin A/C (*Lmna*^+/+^), and controls for the VSMC- and EC-specific models were *Lmna^LCS^*^/*LCS*^ littermates expressing only Lamin C [23]. Specific expression of progerin in VSMCs and ECs was confirmed by immunohistochemistry in aortic sections from *Lmna^LCS^*^/*LCS*^*SM22αCre^tg^*^/+^ and *Lmna^LCS^*^/*LCS*^*Tie2Cre^tg^*^/+^ mice [25].

### 2.2. Nitrite Treatment

When indicated, 6-week-old *Lmna^G609G^*^/*G609G*^ mice were treated for 8 weeks with sodium nitrite (NaNO_2_, Sigma-Aldrich, St. Louis, MO, USA). The compound was dissolved in drinking water at a final concentration of 50 mg/L, a dose that has been reported to be safe in mice, showing no evidence of toxicological or carcinogenic effects and no effect on water consumption [38]. Consistent with these findings, we observed no adverse effects or changes in water consumption in *Lmna^G609G^*^/*G609G*^ or *Lmna*^+/+^ mice treated with sodium nitrite.

### 2.3. Wire Myography

Animals were euthanized at 14–15 weeks of age by CO_2_ inhalation. Immediately after sacrifice, the thoracic and abdominal cavities were opened. Thoracic aortas were excised and immediately placed in ice-cold Krebs–Henseleit solution (KHS: 115 mM NaCl, 2.5 mM CaCl_2_, 4.6 mM KCl, 1.2 mM KH_2_PO4, 1.2 mM MgSO_4_, 25 mM NaHCO_3_, 11.1 mM glucose, and 0.01 mM EDTA), gently cleaned of fat and connective tissue, and cut into ~2 mm long segments. Aortic rings were then mounted on 2 tungsten wires in a wire myograph system (620M, Danish Myo Technology A/S, Hinnerup, Denmark) and immersed in 37 °C KHS with constant gassing (95% O_2_ and 5% CO_2_). Wire myography was performed as previously described [39]. Optimal vessel distension was determined by normalization using the Laplace Equation (Tension = [pressure × radius]/thickness) to calculate the position at which the tension was equivalent to an intraluminal pressure of 100 mmHg (L100) [39]; vessels were then set up to the optimal tension (physiological distension, 0.9 of L100).

After equilibration for 30 min, vasoconstriction was studied by exposing the aortic rings first to 120 mM KCl and then to increasing doses of phenylephrine (from 1 nm to 10 µM; Sigma-Aldrich). We assessed the contribution of vessel stiffness to the contractile function by analyzing contraction induced by 120 mM KCl before and after collagen degradation with collagenase type II (0.2% *w*/*v*, 15 min incubation; Thermo Fisher Scientific, Waltham, MA, USA). Endothelium-dependent vasodilation was assessed by examining the response to increasing doses of acetylcholine (from 0.1 nM to 10 µM; Sigma-Aldrich) in segments previously contracted with 1 µM phenylephrine. Endothelium-independent vasodilation induced by increasing doses of the NO donor diethylamine NONOate (DEA-NO) (from 0.1 nM to 10 µM; Sigma-Aldrich) was examined in segments previously contracted with 1 µM phenylephrine. Drug treatments were separated by extensive washes and a stabilization period of at least 15 min. When indicated, the contribution of NO, Prostacilin I_2_ (PGI_2_), H_2_O_2_, and O_2_^−^ to endothelium-dependent vasodilation was assessed by adding the following agents to the bath 30 min before the acetylcholine dose–response curve: 0.1 mM L-NAME (NOS inhibitor), 10 µM tranylcypromine (PGI_2_ inhibitor), 2000 U/mL catalase (H_2_O_2_ decomposing agent), and 0.1 mM Tempol (O_2_^−^ scavenger) (all from Sigma-Aldrich). All drugs were dissolved in water except for the Tempol stock solution, which was prepared in ethanol.

The results of wire myography experiments are represented as dose–response curves and as area under the curve (AUC), which provides information regarding differences in the dose–response curve as a whole and in a continuous manner.

### 2.4. Statistical Analysis

Results are represented as mean ± standard error of the media (SEM), and statistical differences were analyzed using the tests indicated in the figure legends.

## 3. Results

### 3.1. Impaired Vascular Function in Progeroid Mice with Ubiquitous Progerin Expression

To investigate the impact of ubiquitous progerin expression on vascular function, we performed ex vivo wire myography experiments to examine contraction and dilation in thoracic aorta segments obtained from 14–15-week-old *Lmna^G609G^*^/*G609G*^ mice and age-matched wild-type (*Lmna*^+/+^) littermate controls. These studies demonstrated impaired contraction of progeroid vessel segments in response to incubation with phenylephrine (Figure 1A) and KCl (Figure 1B).

To assess endothelial function, we first exposed phenylephrine-precontrated aortas to the endothelium-dependent vasodilator acetylcholine (Figure 2A). Relaxation induced by the physiological acetylcholine dose (0.1 µM) was significantly lower in *Lmna^G609G^*^/*G609G*^ vessel segments, a result also evidenced by a difference in logEC50 (−7.025 ± 0.06 in *Lmna*^+/+^ aortic rings versus −6.511 ± 0.09 in *Lmna^G609G^*^/*G609G*^ aortic rings) (Figure 2A, left) and a lower AUC for acetylcholine-induced relaxation (Figure 2A, right). In contrast, there were no significant differences in the relaxation of phenylephrine-precontrated *Lmna*^+/+^ and *Lmna^G609G^*^/*G609G*^ aortic rings exposed to the endothelium-independent vasodilator DEA-NO (Figure 2B). These results thus indicate that endothelial dysfunction underlies impaired vessel relaxation in mice with ubiquitous progerin expression.

We next investigated which of the main factors contributing to acetylcholine-induced endothelial relaxation were altered in *Lmna^G609G^*^/*G609G*^ aorta. For this, we prepared acetylcholine-induced relaxation curves in the presence of the following agents: L-NAME (NOS inhibitor), tranylcypromine (PGI_2_ synthase inhibitor), catalase (H_2_O_2_ decomposing agent), and Tempol (O_2_^−^ scavenger). Significant inhibition of acetylcholine-induced relaxation was observed only in L-NAME-treated *Lmna*^+/+^ and *Lmna^G609G^*^/*G609G*^ aortic rings; the other drugs had no significant effect irrespective of mouse genotype (Figure 2C). These findings are consistent with the notion that NO is the main factor underlying endothelium-dependent acetylcholine-induced relaxation in vessels of both genotypes and suggest that NO deficiency may account for the endothelial dysfunction in vessels of mice with ubiquitous progerin expression.

### 3.2. VSMC-Specific Progerin Expression Provokes Contractile Impairment, but Neither VSMC-Specific nor EC-Specific Expression Is Sufficient to Cause Endothelial Dysfunction

VSMCs and ECs are key cellular components of the vessel wall that play major roles in the regulation of vascular tone. To determine the relative contribution of these cell types to impaired vascular tone regulation in progeroid mice, we performed wire myography experiments in aortic rings isolated from *Lmna^LCS^*^/*LCS*^*SM22αCre^tg^*^/+^ and *Lmna^LCS^*^/*LCS*^*Tie2Cre^tg^*^/+^ mice, which express progerin predominantly in VSMCs and ECs, respectively [25]. Controls for both models were *Lmna^LCS^*^/*LCS*^ littermates that do not express progerin [23]. Contraction in response to phenylephrine or KCl was significantly lower in aortic rings from mice with VSMC-specific progerin expression (Figure 3A), like in aortas from *Lmna^G609G^*^/*G609G*^ mice with ubiquous progerin expression (cf. Figure 1A). In contrast, aortas expressing progerin only in ECs contracted normally when incubated with phenylephrine or KCl (Figure 3C). Likewise, vasodilation induced by either acetylcholine or DEA-NO was not significantly different from controls in *Lmna^LCS^*^/*LCS*^*SM22αCre^tg^*^/+^ (Figure 3B) or *Lmna^LCS^*^/*LCS*^*Tie2Cre^tg^*^/+^ aortic rings (Figure 3D). These results demonstrate that smooth muscle, not endothelium, is the main vascular cell type driving contractile impairment in progeroid mice and suggest that progeroid endothelial dysfunction requires simultaneous expression of progerin in VSMCs and ECs.

We next tested whether mechanical impediment by the stiff extracellular cell matrix might contribute to contractile impairment in progeroid mice. Since collagen deposition has been shown to cause aortic stiffness in progeroid *Lmna^G609G^*^/*G609G*^ mice [25], we examined KCl-induced aortic contractions before and after collagen disruption with collagenase. Treatment with collagenase did not improve KCl-induced aortic contraction in *Lmna^G609G^*^/*G609G*^ and *Lmna^LCS^*^/*LCS*^*SM22α^tg^*^/+^ mice (Figure 4A,B).

### 3.3. Treatment with Sodium Nitrite Improves Vascular Function in Progeroid Mice

The key role of NO in progerin-induced endothelial dysfunction (Figure 2C) suggested that NO supplementation might improve vascular tone regulation in progeroid mice. We therefore treated *Lmna^G609G^*^/*G609G*^
*mice* and *Lmna*^+/+^ controls with drinking water supplemented with sodium nitrite (see Materials and Methods). Wire myography with isolated aortic rings showed that nitrite treatment restored sensitivity to acethylcholine-induced aortic relaxation in progeroid aortic rings (Figure 5A, cf. Figure 2A). Nitrite treatment had no effect on DEA-NO-induced endothelium-independent relaxation in progerid or control mice (Figure 5B, cf. Figure 2B). Nitrite treatment also ameliorated the impaired phenylephrine-dependent contraction in aortic rings from progeroid mice (Figure 5C, cf. Figure 1A). In contrast, nitrite treatment did not normalize KCl-induced contraction, which remained significantly lower in progeroid aortic rings control aortic rings than in vessels from *Lmna*^+/+^ controls (Figure 5B, cf. Figure 1B). These results demonstrate that dietary supplementation with sodium nitrite can ameliorate defective vascular tone regulation in progeroid mice.

## 4. Discussion

Aging and associated CVD constitute a major sanitary, societal, and economic challenge [1]. It is therefore of utmost importance to extend knowledge of the cellular and molecular mechanisms underlying vascular aging in order to design more effective diagnostic tools, prevention strategies, and therapies to promote healthier aging. The investigation of vascular aging in humans is challenging due to the difficulty of organizing long follow-up longitudinal studies in large population cohorts, as well as inherent complexity of post-analysis due to the concurrence with aging of other confounding cardiovascular risk factors, such as hypercholesterolemia, diabetes, hypertension, and obesity [2,3,4]. These limitations can be circumvented by studying premature aging syndromes characterized by accelerated CVD but lacking other confounding risk factors [5,6].

The HGPS mouse models generated over the last decade are excellent tools for studying cardiovascular aging over a relatively short time frame and without the presence of confounding cofactors, enabling the identification of the cell types most susceptible to progerin-induced CVD [6,22]. In the present study, we investigated vascular tone regulation in three HGPS models. We found severe contractile impairment in *Lmna^G609G^* mice with ubiquitous progerin expression, which was also observed in *Lmna^LCS^*^/*LCS*^*SM22αCre^tg^*^/+^ with VSMC-specific progerin expression, but not in *Lmna^LCS^*^/*LCS*^*Tie2Cre^tg^*^/+^ mice with EC-specific progerin expression. These findings identify VSMCs as the main cell type targeted by progerin to impair vessel contraction, consistent with our recent studies showing that VSMC-specific progerin expression is sufficient to fully recapitulate vascular alterations observed in mice with ubiquitous progerin expression, including vessel stiffness with inward remodeling; VSMC loss; increased collagen deposition and decreased transverse waving of elastin layers in the medial layer [25]; and accelerated atherosclerosis, medial LDL retention, and plaque vulnerability (when examined in a proatherogenic *Apoe*^−/−^ background) [26,27].

The defective vasoconstriction in progeroid mice reported here is in agreement with results by others showing decreased vasoconstrictor responses in normally aged mice [40,41]. We recently identified collagen deposition by VSMCs as a major contributor to vessel stiffness in *Lmna^G609G^*^/*G609G*^ mice [25]. We therefore hypothesized that progerin-dependent contractile impairment could be due to mechanical impediment by a collagen stiff matrix, smooth muscle cell degeneration, or be caused by a combination of both. Treatment of aortic rings with collagenase to disrupt collagen did not improve contractile responses in *Lmna^G609G^*^/*G609G*^ and *Lmna^LCS^*^/*LCS*^*SM22αCre^tg^*^/+^ aortas, suggesting that collagen deposition is not the cause of impaired vasoconstriction in these animals. VSMC degeneration and dysfunction is therefore the most likely reason for contractile impairment, which might be the initiating cause of progerin-induced collagen deposition and defective vasoconstriction in HGPS models.

Our results show that progeroid endothelial dysfunction requires simultaneous expression of progerin in VSMCs and ECs, since progerin expression in VSMCs or ECs alone does not impair acetylcholine-dependent vessel relaxation. Moreover, vessels from mice with EC-specific progerin expression do not recapitulate the stiffness and inward remodeling observed in mice with ubiquitous or VSMC-specific progerin expression [25]. These findings indicate that the endothelial dysfunction observed in mice with ubiquitous progerin expression must be the result of systemic factors, or that it requires the combination of a dysfunctional medial and endothelial layer. However, since we have used a chemical signal to induce NO-dependent relaxation, we cannot rule out that other mechanically driven signals relevant for endothelial-dependent relaxation might be affected by EC-specific progerin expresion. Interestingly, transgenic mice with EC-specific overexpression of human progerin exhibit interstitial myocardial and perivascular fibrosis without VSMC loss [42]. Excessive collagen production in these mice is primarily derived not from ECs directly but from EC-dependent induction of a profibrotic response in the surrounding tissue [42]. HGPS patients show no alterations to flow mediated dilation [7,11], an indirect measure of endothelial function; nevertheless, the severe atherosclerosis in these patients must be preceded or accompanied by endothelial dysfunction at some point. Indeed, endothelial dysfunction and vascular stiffness might be induced independently but in parallel in HGPS, synergistically promoting vessel dysfunction and atherosclerosis as the disease progresses.

Deficient NO bioavailability is known to cause endothelial dysfunction in various CVD settings and in physiological aging [29,30]. In addition, mice overexpressing human progerin exclusively in ECs have reduced eNOS expression and NO levels [42]. We recently reported that NO supplementation by adding sodium nitrite to drinking water prevents vascular stiffness in progeroid mice [25]. In the present study, we demonstrate that NO is the main driver of endothelial-dependent vasodilation in mice with ubiquitous progerin expression and that treatment with sodium nitrite reverts endothelial dysfunction and partially ameliorates contractile impairment in progeroid mice. Nitrite supplementation increases NO bioavailability without the requirement of L-arginine and NOS [43]. The beneficial effect of sodium nitrite in phenylephrine-induced contractility we observed in progeroid aortic rings might be related to an improvement in VSMC and EC homeostasis rather than to reduced vessel stiffness, since treatment with collagenase to decrease vessel stiffness did not abrogate progerin-induced defects in contractility. Future studies should examine whether the beneficial effect of nitrites on the contraction of progeroid aortic rings can be also explained by local compensatory mechanisms, such as a shift in NO/prostaglandin/EDHF balance.

In summary, our work demonstrates for the first time the presence of vascular tone abnormalities such as a severe VSMC contractile impairment and endothelial dysfunction in a mouse model of premature aging caused by ubiquitous progerin expression. VSMCs are the main cell type involved in this contractile impairment, whereas neither VSMC-specific nor EC-specific progerin expression are sufficient to provoke endothelial dysfunction, which likely requires progerin expression in both ECs and VSMCs, and possibly also in other cell types. Our results also suggest dietary supplementation with nitrites as a novel therapy to treat CVD in HGPS patients.

## Figures and Tables

**Figure 1 cells-09-00656-f001:**
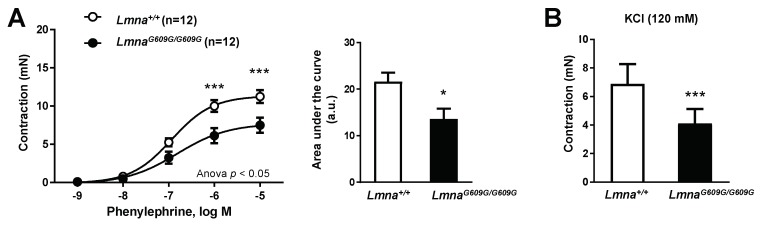
Defective aortic contraction in Lmna^G609G/G609G^ mice. Thoracic aortas from 14-week-old progeroid *Lmna^G609G^*^/*G609G*^ mice (ubiquitous progerin expression) and wild-type (*Lmna*^+/+^) mice were analyzed by wire myography (n = 12 for each genotype). (**A**) Strength of contraction induced by phenylephrine (left) and representation of the area under de curve (right; a.u.: arbitrary units). (**B**) Strength of contraction induced by KCl. Statistical differences were analyzed by two-way ANOVA with Bonferroni’s post-hoc test for phenylephrine data, and by two-tailed t-test for KCl data. * *p* < 0.05 *** *p* < 0.001.

**Figure 2 cells-09-00656-f002:**
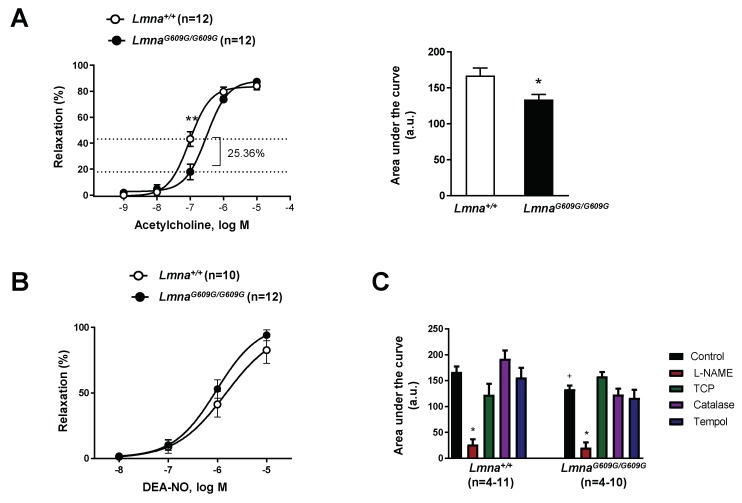
Defective endothelium-dependent aortic dilation in *Lmna^G609G^*^/*G609G*^
*mice.* Thoracic aortas from 14-week-old progeroid *Lmna^G609G^*^/*G609G*^ mice (ubiquitous progerin expression) and wild-type mice (*Lmna*^+/+^) were analyzed by wire myography (*n* = 12 for each genotype). (**A**) Endothelium-dependent vasodilation induced by acetylcholine (left) and representation of the area under the curve (AUC; a.u.: arbitrary units) (right). (**B**) Endothelium-independent vasodilation induced by the NO donor diethylamine NONOate (DEA-NO). (**C**) Representation of area under the curve from acetylcholine-dependent relaxation curves performed in the absence (control) or presence of L-NAME, tranylcypromine (TCP), catalase, or Tempol. Statistical differences were analyzed by two-way ANOVA with Bonferroni’s post-hoc test for acetylcholine and DEA-NO data, and by one-way ANOVA for AUC data. * *p* < 0.05 and ** *p* < 0.01 compared with corresponding control in (**A**) and (**C**); +, *p* < 0.05 *Lmna*^+/+^ versus *Lmna^G609G^*^/*G609G*^ in (**C**).

**Figure 3 cells-09-00656-f003:**
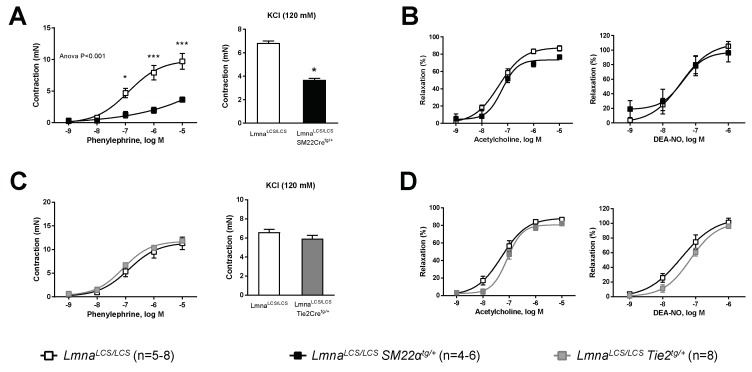
Progerin expression restricted to VSMCs impairs aortic contraction, but VSMC-specific and EC-specific progerin expression is not sufficient to cause defective vessel relaxation. Wire myography was performed in thoracic aorta rings from 14-week-old mice expressing progerin specifically in VSMCs (*Lmna^LCS^*^/*LCS*^*SM22α^tg^*^/+^) or in ECs (*Lmna^LCS^*^/*LCS*^*Tie2^tg^*^/+^) and in control mice without progerin expression (*Lmna^LCS^*^/*LCS*^). (**A**) Contraction induced by phenylephrine (left) and KCl (right) in aortic rings from *Lmna^LCS^*^/*LCS*^*SM22α^tg^*^/+^ mice and *Lmna^LCS^*^/*LCS*^ controls. (**B**) Endothelium-dependent vasodilation induced by acetylcholine (left) and endothelium-independent vasodilation induced by diethylamine NONOate (DEA-NO) (right) in aortic rings from *Lmna^LCS^*^/*LCS*^*SM22α^tg^*^/+^ mice and *Lmna^LCS^*^/*LCS*^ controls. (**C**) Contraction induced by phenylephrine (left) and KCl (right) in aortic rings from *Lmna^LCS^*^/*LCS*^*Tie2^tg^*^/+^ mice and *Lmna^LCS^*^/*LCS*^ controls. (**D**) Endothelium-dependent vasodilation induced by acetylcholine (left) and endothelium-independent vasodilation induced by DEA-NO (right) in aortic rings from *Lmna^LCS^*^/*LCS*^*Tie2^tg^*^/+^ mice and *Lmna^LCS^*^/*LCS*^ controls. Statistical differences were analyzed by two-way ANOVA with Bonferroni’s post-hoc test for acethylcholine, DEA-NO, and phenylephrine data, and by two-tailed t-test for KCl data. * *p* < 0.05; *** *p* < 0.001.

**Figure 4 cells-09-00656-f004:**
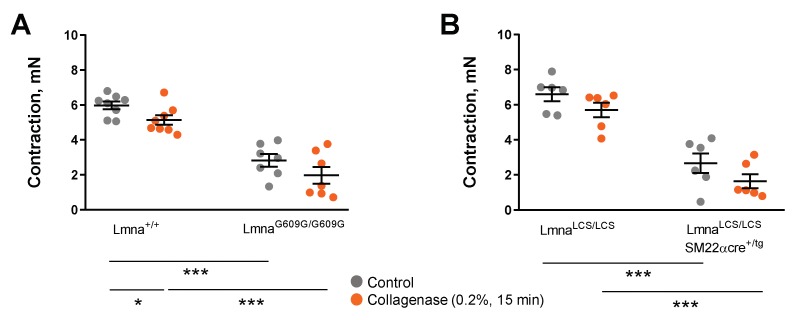
Collagen disruption by collagenase does not rescue contractile impairment in progerin-expressing mice. Wire myography experiments to test the effect of 15 min incubation with 0.2% collagenase on contractile responses induced by 120 mM KCl in aortic rings. (**A**) *Lmna^G609G^*^/*G609G*^ mice and wild-type controls (*Lmna*^+/+^). (**B**) *Lmna^LCS^*^/*LCS*^*SM22α^tg^*^/+^ mice and *Lmna^LCS^*^/*LCS*^ controls. Statistical differences were analyzed with two-way ANOVA followed by the Sidak multiple comparison test. * *p* < 0.05; *** *p* < 0.001.

**Figure 5 cells-09-00656-f005:**
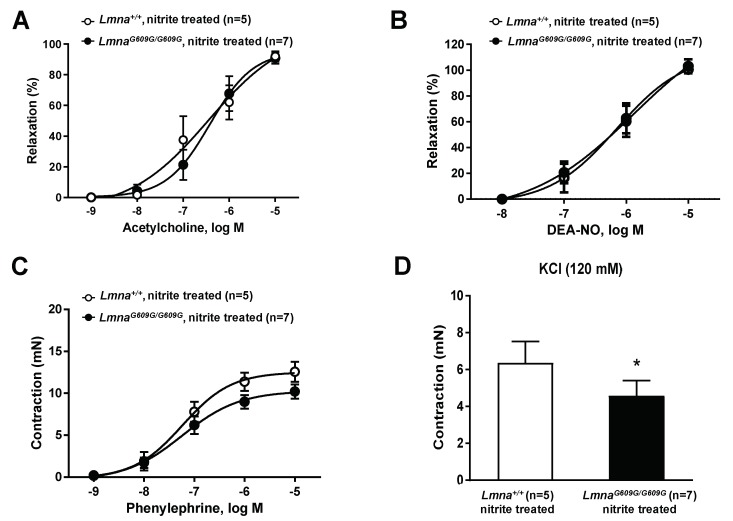
Sodium nitrite treatment improves vascular tone regulation in progeroid mice. Wire myography in thoracic aorta rings from 14-week-old *Lmna^G609G^*^/*G609G*^ and wild-type mice (*Lmna*^+/+^) treated with sodium nitrite in drinking water for 6 weeks. (**A**) Endothelium-dependent vasodilation induced by acetylcholine. (**B**) Endothelium-independent vasodilation induced by diethylamine NONOate (DEA-NO). (**C**) Contraction induced by phenylephrine. (**D**) Contraction induced by KCl. Statistical differences were analyzed by two-way ANOVA with Bonferroni’s post-hoc test for acetylcholine, DEA-NO and phenylephrine data, and by two-tailed *t*-test for KCl data. * *p* < 0.05.

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
