# Peer review of "Vascular Smooth Muscle Cell-Specific Progerin Expression Provokes Contractile Impairment in a Mouse Model of Hutchinson-Gilford Progeria Syndrome that Is Ameliorated by Nitrite Treatment"

_cells, 2020, doi:10.3390/cells9030656_

Round 1

Reviewer 1 Report

cells-725285

The manuscript „cells-725285” entitled: “Vascular smooth muscle cell-specific progerin 2 expression provokes contractile impairment in a 3 mouse model of Hutchinson-Gilford progeria 4 syndrome that is ameliorated by nitrite treatment” by del Campo  L. et al  describes the  endothelial dysfunction and contractile impairment in the aortas of transgenic mice models of accelerated aging.

Three previously described mice models of accelerated aging have been used for the study: lmna \G609G\G609G  expressing progerin, lamin A and lamin C,  lmna\LCS\LCSTie2Cre\tg+ which (as I understand) express progerin mostly with small levels of lamin A ubiquitously and  lmna\LCS\LCSSM22aCretg+ expressing ubiquitously lamin C while progerin expression is restricted to endothelial cells.

The manuscript is a continuation of previous studies on these models in respect to nitrite treatment (del Campo et al 2018; Aging Cell); Hamczyk et al. 2018 Circulation).  

Provided data are interesting and give us better knowledge on the potential mechanisms of development of this type of accelerated aging phenotype in these mice models. However progeric mice model  does not recapitulate all of the human accelerated aging phenotype (as Authors are perfectly aware of (see Introduction).  

The manuscript is very well written all thesis are clearly formulated and discussed.

Authors, in mu opinion should dedicate additional section in Introduction to clearly describe the phenotype of the mice model in respect to lamin proteins  (lamin A, C and progerin). In the present form the manuscript is not entirely clear – especially for general audience interested in aging.  

Reviewer 2 Report

This manuscript highlights the usefulness of genetic aging models to study autonomous senescence of the cardiovascular system. The manuscripts is a good start, but needs to be extended and improved at some points.

It is clear that aging leads to diminished vasonconstriction when looking into isolated blood vessels, but amongst the readership, especially hypertensiologists, geriatrists etc., there is a common, false, assumption that constriction is increased. The manuscript would benefit from an explanation and literature showing decreased constriction (Durik M Circulation 2012, Wu H Clin Sci 2017, Maassen Van Den Brink 1999 Cephalalgia). Is stiffness the reason for decreased cosntriction? Although a possible explanation the results argue against this. Nitrite treatment is only effective for phenylephrine (PE) responses, and not KCl. However, in their previous publication (Del Campo Aging Cell 2019) this group showed effectiveness of nitrite treatment against stiffness. Similarly, constrictions to KCl and angiotensin II are differentially changed in mice lacking proper Ercc1 function (Wu H Clin Sci 2017), which is unexpeted if stiffness is the hampering constrictions. To solve the of stiffness vs. specific phenylephrine-related mechanisms the following could be done: study the effect of proteinases against ECM on constrictions to PE (Del Campo 2019), quantify alpha adrenergic receptors, (responses to) voltage gated a2+ channels, involvement of Rho A kinase, or comparable evidence. What is the mechanism of action of nitrite? Is it NO-cGMP increase which resques the healthy phenotype? Evaluation of the contribution of NO, prostaglandins and EDHF in ACh responses in nitrite treated LmnaG609G/G6096G needs to be performed using specific inhibitors. cGMP levels in plasma and organs in treated vs. untreated animals would be supportive. It is stated on p. 8 that 'endothelial dysfunction observed in mice with ubiquitous progerin expression must be the result of systemic factors'. The lack of decreased ACh and NO-donor responses can however also be explained by local compensatory mechanisms, such as a shift in NO / prostaglandin / EDHF balance. The suggested experiments of point 3 should address this possibility. The best option would of course be to create Sm22alpha/Tie2Cre double Lmna mutants. However, that is a complicated solution. it is surprising the EC-specific Lmna mice show no phenotype. What was the penetrance of the Tie2Cre-lox strategy in these Lmna mutant mice?

Reviewer 3 Report

This study examined the relationship between aging and vascular dysfunction in a mouse model of Hutchinson-Gilford progeria syndrome. The results are quite interesting, because they raise the possibility of developing therapy for the disease. However, there are problems with the control group, and there are some points to be improved in the data and discussion.

Major Problems

  1. This study used LmnaLCS/LCS(G609G) mice as a control group. However, Lamin A is not expressed in this mouse. LmnaLCS/LCS(609 wild type) should be used as a control group.
  2. In Figure 4A, compared to Figure 2A, the error bar appears to be large, and there does not seem to be a significant difference. Therefore, it is doubtful that there was an improvement. In addition, a LmnaG609G/G609G non-nitrite treated group should be used as a comparison in order to clarify the improvement in the nitrite treated group.
  3. Figure 2 shows that NO deficiency in the ECs is one of the underlying causes of vasoconstriction disorders in HGPS. However, the discussion describes VSMSs as the main cause of this contractile impairment. The results should be re-interpreted, and a clearer discussion carried out.
  4. Further discussion is needed as to why progerin expression in VSMCs or ECs alone were not impaired by acetylcholine-dependent vessel relaxation.

Additional minor problems

  1. The meaning of LCS is unclear. Please write "Lmna C-Stop".
  2. The AUC for wire myography was calculated. However, the reason for calculating it is unclear. Please explain the purpose of the calculation, and how it differs from the line graph on the left.
  3. In Figure 2C, you should add the effect of ET-1 inhibitor. Because ET-1 is typical for signal transduction from ECs. Standardize the size and font of the characters that appear in the figures and graphs.
  4. In lines 185 and 208, the mouse line is shown as LmnaG609G / G606G, but I assume that LmnaG609G / G609G is correct.
  5. In Figure 3, the order 3C → 3A → 3D → 3B would be easier to understand, as shown in the order of Figure 1 → Figure 2.
